# Analysis of Cannabinoids Concentration in Cannabis Oil Galenic Preparations: Harmonization between Three Laboratories in Northern Italy

**DOI:** 10.3390/ph14050462

**Published:** 2021-05-14

**Authors:** Alice Palermiti, Alessia Cafaro, Sebastiano Barco, Paolo Bucchioni, Paolo Franceschini, Jessica Cusato, Amedeo De Nicolò, Alessandra Manca, Elisa Delia De Vivo, Eleonora Russo, Francesco Cecchi, Federica Pigliasco, Flavia Lillo, Gino Tripodi, Antonio D’Avolio, Giuliana Cangemi

**Affiliations:** 1Laboratory of Clinical Pharmacology and Pharmacogenetics, Department of Medical Sciences, University of Turin, Amedeo di Savoia Hospital, 10149 Turin, Italy; alice.palermiti@unito.it (A.P.); jessica.cusato@unito.it (J.C.); amedeo.denicolo@unito.it (A.D.N.); alessandra.manca@unito.it (A.M.); elisa.devivo59@gmail.com (E.D.D.V.); 2Chromatography and Mass Spectrometry Section, Central Laboratory of Analysis, IRCCS Istituto Giannina Gaslini, 16147 Genova, Italy; alessiacafaro1996@libero.it (A.C.); sebastianobarco@gaslini.org (S.B.); f.pigliasco@gmail.com (F.P.); ginotripodi@gaslini.org (G.T.); GiulianaCangemi@gaslini.org (G.C.); 3Laboratory of Toxicology, Ospedale San Bartolomeo, ASL5, 19138 Sarzana (SP), Italy; paolo.bucchioni@asl5.liguria.it (P.B.); paolo.franceschini2@asl5.liguria.it (P.F.); 4Department of Pharmacy, University of Genova, 16132 Genoa, Italy; russo@difar.unige.it (E.R.); francesco.cecchi0@gmail.com (F.C.); 5MacroArea dei Laboratori, DIAR dei Servizi A.Li.Sa (Agenzia Ligure Sanitaria), 16100 Genoa, Italy; f.lillo@asl2.liguria.it; 6S.C. Laboratorio di Patologia Clinica—ASL2 Regione Liguria, Department of Diagnostics, 17100 Savona, Italy

**Keywords:** cannabinoids stability, cannabis oil, medical cannabis, UHPLC/MS-MS

## Abstract

Medical cannabis is increasingly being used in the treatment and support of several diseases and syndromes. The quantitative determination of active ingredients (delta-9 tetrahydrocannabinol, THC, and cannabidiol, CBD) in galenic oily preparations is prescribed by law for each produced batch. The aim of this work is to describe the organization of the titration activity centralized at three regional reference laboratories in Northern Italy. Pre-analytical, analytical, and post-analytical phases have been defined in order to guarantee high quality standards. A cross-validation between laboratories allowed for the definition of the procedures that guarantee the interchangeability between reference laboratories. The risk management protocol adopted can be useful for others who need to undertake this activity.

## 1. Introduction

In Italy, *Cannabis sativa* for medical use has been subjected to specific regulation since 2015. Nowadays, in Italy, it is possible to administrate cannabis for therapeutic purposes under medical prescription for some specific uses: in chronic pain and in pain associated with multiple sclerosis and spinal cord injuries; in nausea and vomiting caused by chemotherapy, radiotherapy, and antiretroviral therapies; as an appetite stimulant in cachexia, anorexia, loss of appetite in cancer patients or patients suffering from AIDS, and in anorexia nervosa; for the hypotensive effect in glaucoma; for the reduction of involuntary body and facial movements in the Gilles de la Tourette syndrome (Ministerial Decree 9/11/2015) [1]. 

The Cannabis phytocomplex consists of over 500 substances, including mono- and sesquiterpenes, sugars, hydrocarbons, steroids, flavonoids, nitrogenous compounds and amino acids, among others. The most specific class of Cannabis constituents is the C21 terpenophenolic cannabinoids. The two main cannabinoids at the basis of the therapeutic activity and present in the inflorescences of *Cannabis sativa* L are delta-9 tetrahydrocannabinol (THC) and cannabidiol (CBD) which derive from their acid precursors, tetrahydrocannabinolic acid (THC-A) and cannabidiolic acid (CBD-A), respectively, following oxidation due to exposure to light, air, and direct heating [2]. THC-A and CBD-A derive from a common precursor, cannabigerolic acid (CBG-A). Other cannabinoids present in the phytocomplex are cannabinol (CBN), which originates from the oxidation of THC, cannabichromene (CBC) and cannabigerol (CBG) [3]. The combination of the two cannabinoids THC and CBD with the other components of the phytocomplex is the basis of the efficacy of medicinal extracts. Moreover, many authors suggest that the low incidence of side effects, compared to synthetic drugs with a similar therapeutic indication, are related to the mixture of bio-substances present in cannabis plant extracts [4].

THC and CBD have distinct pharmacological properties: THC has a predominantly psychotropic activity whereas CBD has analgesic and antioxidant activity and is able to mitigate THC side effects [5,6,7,8]. In particular, preparations with a higher CBD content are indicated for epilepsy, inflammation, psychosis or mental disorders, inflammatory bowel disease, migraine, depression, and anxiety, whereas preparations with a higher THC content for pain, muscle spasticity, glaucoma, insomnia, low appetite, cachexia, nausea due to chemotherapy, and anxiety [9,10]. To date, Sativex^®^ is the only cannabis-based herbal medicine authorized for the Italian market. It consists of a spray formulation of a mixture of two cannabis sativa extracts, with CBD and THC. Sativex^®^ is only indicated in patients with moderate to severe spasticity that have responded inadequately to other anti-spasticity medication. Recently, a highly purified CBD oral solution (Epidyolex^®^) has been approved by European Medicines Agency (EMA) [9]. It is indicated as an adjunct therapy, in combination with clobazam, for seizures associated with Lennox–Gastaut syndrome (LGS) or Dravet syndrome in patients aged > 2 years. At the moment, in Italy, it is allowed for compassionate use only and its authorization is currently under evaluation at the Italian Medicines Agency (AIFA). The Decree of the Ministry of Health of 9 November 2015 authorizes the Military Chemical and Pharmaceutical Works of Florence (ICFM) to the autonomous cultivation/production of medical cannabis. The first batches of the product called Cannabis FM2, with standardized concentrations of THC (5–8% *w*/*w*) and CBD (7.5–12% *w*/*w*), have been available since January 2017 and a second product, called FM1, with a declared THC content of 13–20% *w/w* and CBD <1% *w/w*, since July 2018. FM1 and FM2 are distributed to authorized pharmacies for the preparation of magisterial galenic preparations. FM1, having a higher THC/CBD ratio is prescribed for conditions treated with an higher concentration of THC; on the contrary FM2, with a higher CBD/THC ratio, is prescribed for conditions treated with a higher concentration of CBD. The therapeutic indications are decided case by case by the physician. The production of Cannabis FM1 and FM2 by ICFM takes into account the consumption of the last two years and its annual increase [11]. 

Data from the Ministry of Health show a progressive growth in national cannabis use which in 2019 amounted to about 860 tons of raw material. To meet the growing demand, the production is integrated with the import of non-registered Dutch Cannabis-based products, in relation to the willingness to export of the Office of Medicinal Cannabis, which applies the directives of the Ministry of Health, Welfare and Sport Dutch on the export of such products [12]. Cannabis varieties imported in Italy from Holland are: Bedrocan (average THC concentrations 22% *w/w* and CBD <1%), Bedrobinol (13.5% THC and <1% CBD), Bediol (6.5% THC and 8% for CBD) and Bedrolite (0.4% THC and 9% CBD *w/w*). The Liguria Region, together with Tuscany, Emilia Romagna, Puglia, Lombardy, and Piedmont, is among the regions with the greatest use of cannabis. In December 2019, the number of treated patients was estimated at 1056 in the region, most of them adults. The recommended methods of administering medical cannabis are decoctions and oily extract in drops, and more rarely smoking or vaporization, with preparation methods established nationally. It has been shown, however, that the recovery of the different active ingredients in the decoction is very limited and with variable ratios between THC and CBD [13,14,15]. Furthermore, the stability of cannabinoids in aqueous solution is very low and this makes it possible to use only extemporaneous preparations [16]. Oily preparations, on the other hand, are characterized by a higher THC/CBD ratio, resulting in a better recovery of THC and a greater stability of the preparation over time, with a decrease in the concentration of the active compounds of less than 20% in the first days of preparation and a substantial long-term stability and up to one year at ambient temperature. The different methods for the preparation of galenic preparations are reported in the scientific literature, however the absence of standardization seems to be the basis of the considerable variability in the concentrations of active ingredients that are found [17,18]. The oily galenic preparations are produced in Liguria Region by the hospital pharmacies of the Public Health Companies (ASL2, ASL3, ASL4 and IRCCS Giannina Gaslini Institute) and by some private pharmacies in compliance with the “Technical Standards approved by the Ligurian Health Authority”, approved by ALISA (Azienda Ligure Sanitaria, Ligurian Health Authority) resolution 78 of 2018 [19]. In some countries, including Italy, the titration of active ingredients is required for each batch of cannabis oil produced. In fact, based on the content of carboxylated and decarboxylated cannabinoids, decoctions and cannabis oil have shown different pharmacokinetic properties in treated individuals [20]. The oily galenic cannabis preparations must be titrated in order to ensure the quality of the product and to allow the correct setting of the therapeutic protocol. This means that the determination of the concentration of the active ingredient(s) for each preparation batch should be performed with sensitive and specific methodologies, such as liquid or gas chromatography coupled with mass spectrometry. The galenic preparations prepared in Liguria Region have been sent for titration to an external certified reference laboratory (Laboratory of Clinical Pharmacology and Pharmacogenetics, on behalf of CoQua Lab srl, Turin) (laboratory 1) until December 2019. Given the high number of patients treated with cannabis derivatives in the Liguria region, two regional laboratories (the Chromatography and Mass Spectrometry Section of the Central Laboratory of Analysis of the IRCCS Giannina Gaslini Institute (Genoa) (laboratory 2) and Toxicology Laboratory of Sarzana (La Spezia), (laboratory 3) have been identified as regional reference for the titration activity. The aim of this work is the description of the protocol of process management and harmonization for the titration activity of the three reference laboratories.

## 2. Results

### 2.1. Pre-Analytical Phase

The methods of packaging, transport, and acceptance have been defined in detail by a standard operating procedure in compliance with ISO 9001/2015 Quality Management System [21]. The cannabis-based oily galenic preparation, prepared according to the A.Li.Sa technical standards [19], must be mixed and homogenized with careful mechanical stirring immediately after preparation, according to the packaging procedures. Then, 0.2 mL aliquot of each preparation must be taken into an amber glass vial and stored at 2–8 °C until delivery to the reference laboratories, using temperature and path tracking systems and respecting the chain of custody. In each reference laboratory, the acceptance of samples takes place using the internal laboratory informative systems (LIS), reporting the identification of the sending pharmacy, the batch number, and production date of the galenic preparation.

### 2.2. Analytical Phase

The three reference laboratories developed and validated single analytical methods for the quantitative determination of THC, CBD, THC-A, and CBD-A by liquid chromatography-tandem mass spectrometry (LC-MS/MS) using different instruments, sample preparation methods, and chromatographic conditions (see Section 4.3).

The methods were specific with the absence of interfering peaks at the specified LC-MS/MS conditions (4.3). Linearity from 5 ng/mL (lower limit of quantification, LLOQ) to 2000 ng/mL (upper limit of quantification, ULOQ) for THC, CBD, THCA, and CBDA was obtained (102–25,000 ng/mL Lab1). Figure 1 shows representative chromatograms.

A linear relationship between the analyte’s peak area and the corresponding concentration in the entire concentration range (with R^2^ > 0.98) for all the analytes was obtained for all the methods. The back-calculated concentration values for all the analytes were within ±15% of the theoretical value. Results of the intra- and inter-assay precision and accuracy and recoveries were all <15%. A carry-over effect wasn’t detected, and the sample dilution had no effect on the analyte’s determination. The matrix effect was negligible according to the very high dilution of the oil in the solvent.

### 2.3. Cross Validation and Harmonization Between Laboratories

A cross validation of the analytical methods applied in the three laboratories was carried out. First, a set of calibrators and quality controls (QCs) were prepared in each laboratory and sent to the other two. The results obtained were highly comparable, with coefficients of variation (CV) <10%. Then, galenic preparations were produced by a single hospital pharmacy and sent in triplicate to the three reference laboratories. In this case, the results showed a high degree of variability (>20%) between the three laboratories in the measured concentrations of the four cannabinoids (data not shown).

All the possible sources of variability were then analyzed in order to harmonize the results between the three laboratories. The main source of variability was identified in the sample preparation protocol.

Thus, a single batch of FM2 oily galenic preparation was analyzed using different dilution schemes (as described in Section 4.5.1) in order to determine which procedure guaranteed higher precision. The CV%, calculated on the average of five repetitions, obtained with the four methods are described in Table 1 for pipettes set 1 (standard Gilson PIPETMAN^®^ Classic^®^) and in Table 2 for pipettes set 2 (Gilson MICROMAN^®^ for viscous liquids).

Noticeably, the CV% increased when schemes with a higher number of operator’s steps were adopted. The scheme that guaranteed lower CV% was obtained with Gilson MICROMAN^®^ for viscous liquids with samples transferred through a single pressure of the plunger into the solvent.

Consequently, the three laboratories adopted the same sample preparation protocol (method 2 with pipettes for viscous fluids described in Section 4.5.1). A cross-validation was then performed analyzing 23 galenic oily samples prepared and sent in triplicate by the hospital pharmacies of ASL2, ASL4, and IGG.

The obtained results are shown in Figure 2.

The percentage difference between the results of the three laboratories were within 20% for at least 67% of the determinations, according to EMA guidelines.

Passing–Bablok correlation plots of the cannabinoid concentrations, obtained in the three laboratories are shown in Figure 3.

Table 3 shows Pearson’s correlation coefficients slope and intercepts with 95% confidence intervals (CI) obtained.

According to the Cusum test for both THC and CBD for all the three laboratories, there was no significant deviation from linearity (*p* > 0.1). Results of the Passing–Bablok test (Table 3) show that both systematic and proportional differences were absent except for a very slight proportional difference in THC measurements between lab 2 and lab 3.

Standard operating procedures (Standard Operating Procedures, SOPs) were drawn up for each laboratory in compliance with the respective quality systems in force (ISO, Joint Commission International

### 2.4. Stability of Cannabinoids in FM2 Galenic Preparations

Results of stability are shown in Figure 4.

The four cannabinoids were stable within 27 days in the FM2 preparation both at room temperature and at 4 ± 3 °C, with percentage differences always inside the acceptable ranges.

### 2.5. Post-Analytical Phase and External Quality Assessment

Common procedures for the reports and turnaround times have been established. The reports include the concentrations of the four cannabinoids (expressed in mg/mL), the identification of the production pharmacy, the batch number and the date of preparation of the galenic, the date of acceptance in the laboratory system and the analytical method used (LC-M/MS). The report is digitally signed and sent by email within 48 h from receipt of the sample. In December 2019, the three laboratories participated to an External Quality Assessment (VEQ) program for the titration of cannabinoids in cannabis oil preparations organized by the National Center for Addiction and Doping of the Istituto Superiore di Sanità with results within the ranges of acceptance for all the analytes.

## 3. Discussion

The determination of the exact cannabinoid content in galenic preparations is crucial to correctly guide clinicians in prescribing therapy and for pharmacists to assess and ensure the quality of the preparation. Growing evidence regarding the correlations between THC–CBD blood concentrations and the administered doses support the need for the therapeutic drug monitoring of cannabis, highlighting the importance of a reliable titration of preparations [14]. A high variability of the content of magistral preparations has been described, underlying the need for a uniquely defined protocol to be adopted by all galenic laboratories which guarantee reproducible and controlled extraction yields [18]. On the other hand, a uniquely defined protocol might not be useful considering all the possible combinations of galenic preparations, with different percentages of active substances.

In this paper, we have shown for the first time the process management of the titration activity centralized in three reference laboratories in Northern Italy. The aim of our project was to standardize the entire process from the galenic preparations to the titration activity. Pre-analytical, analytical, and post-analytical phases have been standardized in order to ensure a high quality of the results, short turnaround times, and cost-effectiveness. In Italy, the titrations for each batch of galenic product should be performed, preferably by mass spectrometry either coupled with gas or liquid chromatography. Nevertheless, gas chromatography (GC) does not allow for the direct analysis of the extracted sample requiring a derivatization step and a heating of the sample at very high temperatures which produce a thermal conversion of the cannabinoid acids in their neutral derivatives in the GC injection port. On the contrary, LC-MS/MS, which has become the gold standard for the quantitative determination of cannabinoids in different matrices, allowed for analysis at room temperature, thus guaranteeing a correct quantification of both neutral and acid forms of THC and CBD, which is an important prerequisite for dosage determination [20].

Several sample preparation protocols have been previously described in literature adopting different solvents such as isopropanol [18], methanol: chloroform [11], tetrahydrofuran [22], ethanol. Both liquid-liquid extractions and dilute-and-shoot methods were adopted. All these methods ensured a high recovery of the cannabinoids from the oily matrix. In our protocol, we have adopted the simple dilute-and-shoot procedure with isopropanol described by Carcieri et al. [18]. The use of deuterated internal standards guarantees method robustness.

The analytical methods were validated in each laboratory using QCs obtained by spiking cannabinoids in surrogate matrix and a first cross-validation between laboratories with the same QCs gave excellent results. Nevertheless, when cross-validation was performed using galenic preparations, a high variability was evident, making the adoption of harmonization strategies necessary. Different sample preparation procedures were then tested and the method able to ensure lower CV% (using pipettes for viscous liquids with a low number of steps) was adopted. Despite the different sample preparations and the different instruments and analytical methods, the results obtained were highly comparable. Of note, the method with the weighing of the sample gave acceptable CV%, but was less accurate because it was based on the density of pure olive oil instead of cannabis oil. This last procedure can be recommended in the case of an unavailability of pipettes for viscous fluids. The standardization of the LC-MS/MS methods adopted allowed us to obtain interchangeable results and the consequent possibility to alternate the titration activity between the clinical laboratories. Moreover, the stability of the galenic preparations over a one-month period ensures that the concentration of THC and CBD remains stable within the expiry defined by the master’s receipt duration, confirming findings by other groups [15,16]. Pacifici et al. demonstrated that both short (two weeks) and long (one-year) term storage of FM2 at ambient temperature caused only slight changes in the content of cannabinoids.

## 4. Materials and Methods

### 4.1. Chemicals and Reagents

HPLC-grade acetonitrile (ACN) was purchased from VWR Chemicals (Radnor, PA, USA); MS-grade water (MilliQ) was produced with a Milli-DI system coupled with a Synergy 185 system by Millipore (Milan, Italy); Formic acid (99.9%) and isopropyl alcohol were purchased from Merck (Darmstadt, Germany), pharmaceutical grade olive oil was purchased from Sigma Aldrich (Milan, Italy) or, alternatively, Olive refined European Pharmacopeia oil (Olive Oil, Ph.Eur) was purchased from A.C.E.F. (Fiorenzuola d’Arda, Pisa, Italy). Δ9-THC 1.0 mg/mL in methanol solution (T-005 Cerilliant), Δ9-THC-D3 100 μg/mL in methanol solution (T-003 Cerilliant; used as THC internal standard (IS)), CBD 1.0 mg/mL in methanol solution (C-045 Cerilliant), CBD-D3 100 μg/mL in methanol solution (C-084 Cerilliant; used as CBD IS), THC-A 1.0 mg/mL in acetonitrile (T-093 Cerilliant) and CBD-A 1.0 mg/mL in acetonitrile (C-144 Cerilliant) were purchased from Sigma-Aldrich Srl (Milan, Italy). Δ 9-THC-D3 purity was >96% while all the other compounds had purity >98%. All standards were stored a −80 °C in the dark, in order to prevent any possible degradation.

### 4.2. Preparation of Calibrators and Quality Control Samples

#### 4.2.1. Lab1 preparation

Surrogate matrix was obtained by diluting 1:20,000 olive oil in isopropanol. Calibrators and quality controls were prepared by diluting a working solution of analytes with appropriate surrogate matrix volume. The calibration curve, for all compounds, was in the range 5.14, 1250 ng/mL and it was composed by seven points (0, 5.14, 15.4, 46.3, 138.8, 416.6, 1250 ng/mL). The four QCs concentrations were 12.5 ng/mL (LLOQ), 50 ng/mL (QC-L), 500 ng/mL (QC-M), and 1000 ng/mL (QC-H). Internal standard working solution (IS) was prepared with CBD-d3 and THC-d3 at 1.4 µg/mL in 50:50 (water: methanol).

#### 4.2.2. Lab2 and Lab3 Preparation

Surrogate matrix was obtained by diluting 1:10,000 olive oil in isopropanol. Calibrators and quality controls were prepared by diluting a working solution of analytes with appropriate surrogate matrix volume. The calibration curve, for all compounds, was in the range 5–2000 ng/mL and it was composed by twelve points (5, 25, 50, 100, 200, 400, 600, 800, 1000, 1500, and 2000 ng/mL). The four QCs concentrations were 5 ng/mL (LLOQ), 80 ng/mL (QC-L), 300 ng/mL (QC-M), and 900 ng/mL (QC-H). Internal standard working solution (IS) was prepared with CBD-d3 and THC-d3 at 0.2 µg/mL in 50:50 (water:methanol).

### 4.3. Sample Preparation

#### 4.3.1. Lab1 Sample Preparation

The galenic oil sample was prepared by serial dilution 1:10,000 in isopropanol and addition of the deuterated internal standards (CBD-d3 and THC-d3). A 50 µL aliquot of galenic oil sample was 10,000-fold diluted with isopropanol in two steps, vortex mixing each time. A 70 µL aliquot of diluted galenic oil sample, calibrator or QCs was added to 70 µL internal standard working solution directly in autosampler vials, for an overall 20,000-fold dilution, vortex mixed, and injected in the LC-MS system.

#### 4.3.2. Lab2 and Lab3 Sample Preparation

The galenic oil sample was prepared by serial dilution 1:10,000 in isopropanol and addition of the deuterated internal standards (CBD-d3 and THC-d3). A 10-µL aliquot of galenic oil sample was 10,000-fold diluted with isopropanol in two steps, vortex mixing each time. A 25 µL aliquot of diluted galenic oil sample, calibrator or QCs was added to 25 µL internal standard working solution directly in autosampler vials, for an overall 20,000-fold dilution, vortex mixed, and injected in the LC-MS system.

### 4.4. LC-MS Analysis

#### 4.4.1. Lab1 LC-MS Analysis

LC-MS/MS analyses were performed on an Acquity^®^ UPLC (Waters, Milan, Italy), consisting of a binary pump, a refrigerated sample manager and a triple quadrupole detector (TQD, MS/MS). Chromatographic separations were carried out on an Acquity UPLC HSS-T3 2.1 × 30 mm, 1.8 µm column. The adopted chromatographic conditions were as follows. Gradient separation chromatography was carried out with mobile phase A consisting of 0.05% formic acid in water:acetonitrile (30:70), and mobile phase B of 0.05% formic acid in isopropanol:acetonitrile (80:20). The percentage of A phase started at 100% for 4.6 min, programmed to reach 100% of B phase and kept for 1.5 min, then the column was reconditioned at 100% B for a total run time of 7 min. The flow rate was 0.4 mL/min. Injection volume was 4 µL and total run time was 7 min. The separated analytes were detected with a triple quadrupole mass spectrometer operating in multiple reaction monitoring (MRM) mode via positive electrospray ionization (ESI). The applied ESI conditions were the following: capillary voltage 3.0 kV, Ion Transfer Tube temperature and Vaporizer temperature were 150 °C and 400 °C respectively. Ionization was achieved using electrospray ionization (ESI) in the positive ion mode and analytes were detected using selective reaction monitoring (SRM) of the specific following ion transitions: for THC and CBD [M+H] + 315.2→193.1 and 259.2 m/z and their deuterated IS [M+H] + 318.2 → 196.1 and 296.15 m/z, for CBD-A 359.15→261.1 m/z and THC-A 341.15→219.15 m/z with positive ionization.

#### 4.4.2. Lab2 Lab3 LC-MS Analysis

LC-MS/MS analyses were performed on two different instruments: (1) 6430 Triple Quad coupled to an Agilent 1200 series HPLC (Agilent, Milan, Italy) (Lab 2) and (2) Acquity UPLC system coupled with a Triple Quadrupole Detector (TQD, MS/MS) (Lab 3). Chromatographic separations were carried out on a Zorbax Eclipse Plus C18 2.1 × 50 mm, 1.8 µm. The adopted chromatographic conditions for the two laboratories were the following: gradient separation chromatography was carried out with mobile phase A consisting of 0.1% formic acid in water, and mobile phase B of 0.1% formic acid in acetonitrile. The percentage of solvent B started at 50% for 0.2 min, programmed to reach 100% in 2.8 min and kept for 2 min, then the column was reconditioned at 50% B for a total run time of 8.5 min. The flow rate was 0.3 mL/min. Injection volume was 5 µL and total run time was 8.5 min. The separated analytes were detected with a triple quadrupole mass spectrometer operating in multiple reaction monitoring (MRM) mode via positive electrospray ionization (ESI). The applied ESI conditions were the following: capillary voltage 3.0 kV, Ion Transfer Tube temperature and Vaporizer temperature were 300 °C. Nitrogen was used as the nebulizer and auxiliary gas, set at 40 and 10 arbitrary units, respectively. For collision-induced dissociation, high purity argon was used at a pressure of 1.5 mTorr with a collision energy of 25 V. Q1 and Q3 resolution was 0.2 and 0.4 full-width–half-maximum (FWHM), respectively. Ionization was achieved using electrospray ionization (ESI) in the positive ion mode and analytes were detected using selective reaction monitoring (SRM) of the specific following ion transitions: for THC and CBD [M+H] + 315.2 → 123, 135.1, 193.1 m/z and their deuterated IS [M+H] + 318.2, 262→196.1 m/z, for CBD-A and THC-A 357.2→245.1 with negative ionization.

### 4.5. Method Validation

Method validation was carried out in each laboratory following guidelines for the validation of a quantitative chromatographic bioanalytical method [21,23]. Selectivity, carry over, matrix effects, intra- and inter-assay reproducibility, and accuracy were evaluated on QC samples.

#### 4.5.1. Analysis of Precision with Oily Galenic Samples

Four different sample preparation protocols were tested by using two different sets of pipettes: (set 1: standard Gilson PIPETMAN^®^ Classic^®^; set 2: Gilson MICROMAN^®^ for viscous liquids) by the same operator on the same day on the same batch of FM2 oily galenic preparation.

(1)A 10-µL aliquot of galenic preparation was transferred in an amber glass autosampler vial and weighted on an analytical balance. The correct volume was then calculated using the density of pure olive oil.(2)A 10-µL aliquot of galenic preparation was added to 990 µL isopropanol in an amber glass autosampler vial by immersing the tip in the solvent and transferring the oil with a single push of the plunger.(3)A 10-µL aliquot of galenic preparation was added to 990 µL of isopropanol in an amber glass autosampler vial by immersing the tip in the solvent and transferring the oil with a single push of the plunger, after cleaning the outer walls of the tip with absorbent paper in order to eliminate any residue of oil,(4)A 10-µL aliquot of galenic preparation was added to 990 µL of isopropanol previously inserted in vials by immersing the tip in the solvent and transferring the oil with three pushes of the plunger in order to recover the greatest amount of oil left inside the tip.

After the first dilution, second dilution in isopropanol was applied for all protocols by adding a 10-µL aliquot from the first dilution to 990 µL isopropanol. Five repetitions were done for each method. All analyses were performed on the same day and by the same operator.

#### 4.5.2. Cross Validation and Harmonization between Laboratories

A total of 23 samples (3 Bedrolite, 7 Bediol, 5 Bedrocan, 8 FM2) obtained in triplicate from the galenic preparations produced by ASL2 (no.12), ASL4 (no.5) and IRCCS Istituto Giannina Gaslini (no.6) were sent in triplicate to the three laboratories using the same delivery procedures and analyzed using the same sample preparation procedure. The results obtained were compared following the cross validation acceptance criteria described by EMA guidelines for bioanalytical method validation calculating the percentage difference between the three values obtained [24]. Results were considered acceptable within 20% of the mean for at least 67% of the repeat. Moreover, Passing-Bablok linear regression was also evaluated using MedCalc software (MedCalc Software Ltd., Ostend, Belgium).

#### 4.5.3. Analysis of Stability

The stability of cannabinoids was assessed on a single batch of FM2 stored at room temperature (25 ± 2 °C) or at 4 ± 3 °C in the dark analyzed in triplicates at the following timepoints: 5, 7, 14, 20 and 27 days. As suggested by EMA guidelines, samples were considered stable if the percentage difference, calculated as the ratio between the concentration measured at each sampling point and the initial concentration, was lower than 15% [24].

## 5. Conclusions

The quantitative determination of cannabinoids in oily galenic preparations is crucial to ensure an appropriate dosage in therapeutic cannabis administration. In this paper, we have described a risk management protocol adopted by three laboratories in Northern Italy that can be used as a model for other regional projects and can be exported to other extra-regional contexts.

## Figures and Tables

**Figure 1 pharmaceuticals-14-00462-f001:**
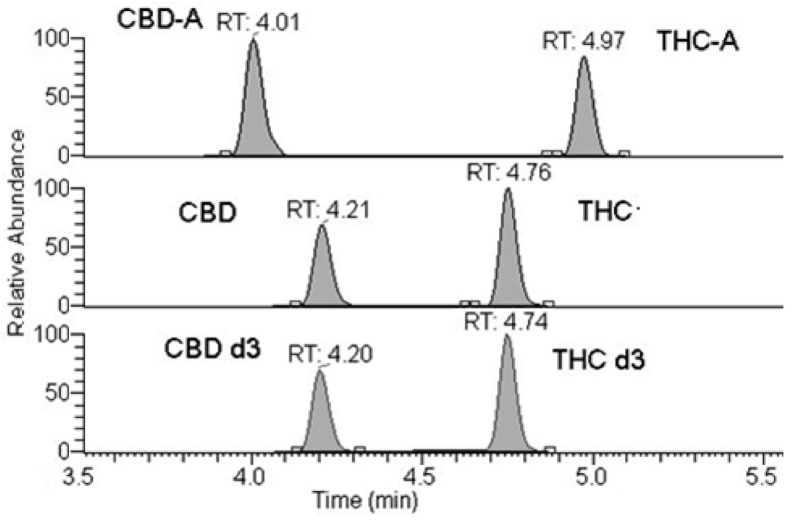
Chromatographic peaks and retention times of CBD, CBD-A, THC and THC-A and deuterated internal standards.

**Figure 2 pharmaceuticals-14-00462-f002:**
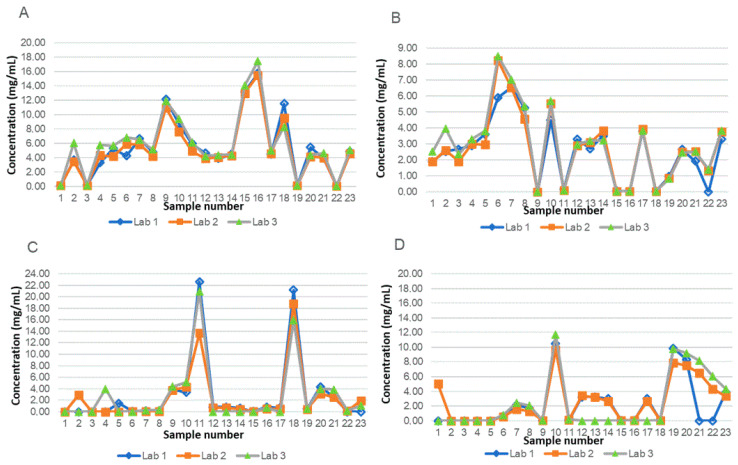
Concentration (mg/mL) of THC (**A**), CBD (**B**), THC-A (**C**) and CBD-A (**D**) measured on 23 galenic preparations in the three laboratories during cross-validation.

**Figure 3 pharmaceuticals-14-00462-f003:**
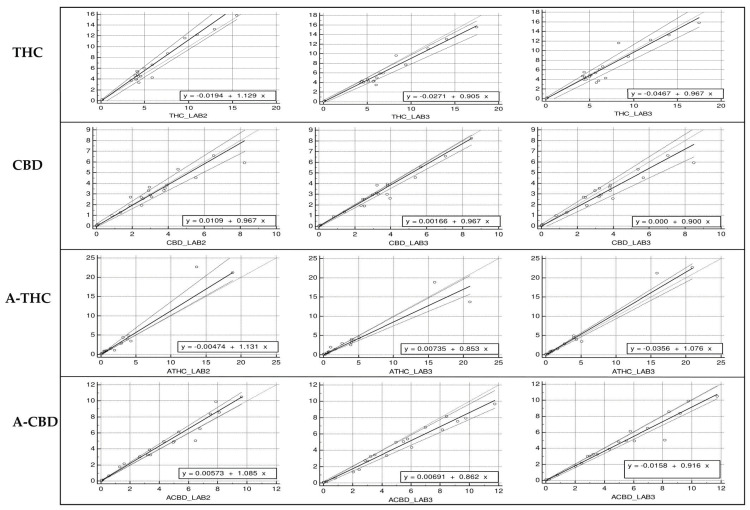
Passing Bablok correlation plots for THC, CBD, A–THC and A–CBD. The plots show excellent correlations between the three laboratories for both cannabinoids.

**Figure 4 pharmaceuticals-14-00462-f004:**
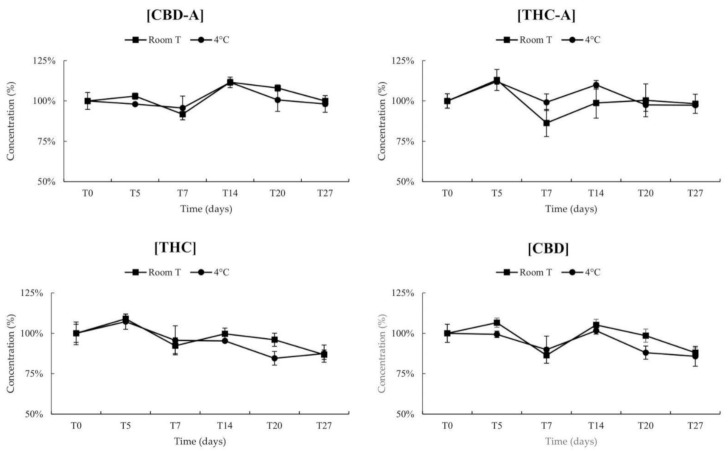
Stability of cannabinoids in FM2 at different temperatures and up to 27 days. The results are expressed as % initial concentration mean of three replicates. Bars indicate CV%.

**Table 1 pharmaceuticals-14-00462-t001:** (CV%) of the different dilution schemes, with pipettes 1.

Method	THC CV%	CBD CV%	A-THC CV%	A-CBD CV%
**1**	5.9	1.0	22.7	8.2
**2**	21.8	23.3	19.9	18.1
**3**	29.1	26.5	28.8	27.6
**4**	20.7	21.7	26.1	23.6

**Table 2 pharmaceuticals-14-00462-t002:** (CV%) of the different dilution schemes, with pipettes 2.

Method	THC CV%	CBD CV%	A-THC CV%	A-CBD CV%
**1**	7.4	14.5	17.7	7.4
**2**	6.7	3.7	6.5	4.0
**3**	15.0	12.4	11.5	11.1
**4**	9.3	9.7	10.1	6.6

**Table 3 pharmaceuticals-14-00462-t003:** Results of the Passing Bablok method comparison.

		Correlation Coefficient r (95% CI)	Linearity (*p* Value by Cusum Test)	Slope B (95% CI)	Intercept A (95% CI)
**LAB1–LAB2**	**THC**	0.9844 (0.9630–0.9935)	0.72	1.1292 (1.0140–1.2505)	−0.0194 (−0.6777–0.1399)
	**CBD**	0.9597 (0.9059–0.9830)	0.78	0.9668 (0.8567–1.0607)	0.0109 (−0.0848–0.2247)
	**A-THC**	0.9736 (0.9364–0.9892)	0.42	1.1313 (1.0217–1.3566)	−0.0047 (-0.0862–0.0528)
	**A-CBD**	0.9830 (0.9597–0.9929)	0.45	1.0854 (0.9985–1.1429)	0.0057 (-0.0529-0.0608)
**LAB2–LAB3**	**THC**	0.9836 (0.9610–0.9931)	0.42	0.9032 (0.8156–0.9370)	−0.0155 (-0.1749–0.3029)
	**CBD**	0.9845 (0.9632–0.9935)	0.72	0.9667 (0.9052–1.0000)	0.0017 (−0.0700–0.0546)
	**A-THC**	0.9481 (0.8797–0.9781)	0.78	0.8529 (0.7565–0.9793)	0.0074 (−0.0928–0.0773)
	**A-CBD**	0.9882 (0.9718–0.9951)	0.66	0.8619 (0.7912–0.9514)	0.0069 (−0.1492–0.1625)
**LAB1–LAB3**	**THC**	0.9592 (0.9047–0.9828)	0.78	0.9666 (0.9010–1.1029)	−0.0467 (−0.8512–0.1831)
	**CBD**	0.9653 (0.9185–0.9854)	0.99	0.9003 (0.7912–1.0539)	0.0000 (−0.2466–0.1625)
	**A-THC**	0.9891 (0.9735–0.9956)	0.78	1.0756 (0.9487–1.1290)	−0.0356 (−0.1255–0.0244
	**A-CBD**	0.9800 (0.9526–0.9916)	0.78	0.9157 (0.8808–1.0000)	−0.0158 (−0.1700–0.0557)

## Data Availability

Data will be provided on request.

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
