# Peer review of "Analysis of Cannabinoids Concentration in Cannabis Oil Galenic Preparations: Harmonization between Three Laboratories in Northern Italy"

_pharmaceuticals, 2021, doi:10.3390/ph14050462_

Round 1

Reviewer 1 Report

This manuscript, entitled “Analysis of cannabinoids concentration in cannabis oil galenic preparations: harmonization between three Laboratories in Northern Italy” describes the titration of the analysis of cannabis medical preparation. This is an interesting research area, as the use of cannabis is increasing. It is suitable for publication in pharmaceuticals. However, many aspects should be reviewed before the publication. Here there are some comments and suggestions.

Introduction:

  1. In the first paragraph the authors described the medicinal uses of Cannabis sativa in Italy. What is the legal situation in Italy? Are there plant extracts (Tilray, Bedrocan….) allowed and prescribed by doctors?
  2. Are there isolated cannabinoid formulations approved (Sativex, Canemes,..). At least Sativex is approved by EMA and marketed in several EU countries. Epidiolex is under investigation. I suggest including this information. Moreover, Is there any restriction with the THC content of these extracts?
  3. It should be considered that not all cannabis extracts are useful for “everything”, some effects are attributed to CBD. Others to THC.  (CBD epilepsy, anxiety…; THC: antiemetic in patients receiving chemotherapy, anorexia, pain, spasticity). This should be considered when explaining the benefit of Cannabis.
  4. At the end of the third paragraph. It is true than in general, except the psychotropic side effects related to this plant, cannabis is well tolerated. THC is the main responsible for cannabis psychedelic activity and CBD palliates the psychotropic side effects of THC. However, apart from this they show, in general, a similar safety profile (dry mouth, dizziness, drowsiness are the most common adverse events related to both CBD and THC treatments). In general, synthetic CBD and THC are not more toxic than the extracted one. The differences can be related to the administered amount and in case of THC psychotropic side effects, the combination with CBD. What did the authors mean in the sentences “ The combination of the two cannabinoids THC and CBD with the other components of the phytocomplex is at the basis of the efficacy of medicinal extracts and the low incidence of side effects compared to synthetic drugs with a similar therapeutic indication ?
  5. FM1 and FM2 preparations. The authors should include the therapeutic indication.

Methods:

  • IS should be defined in the section 4.1
  • Why lab 1 used different concentrations for calibration curves?
  • In section 4.5.3 Did the authors use positive control? I guess than the authors follow ICH guidelines for stability studies. The reference should be included.
  • Where did the author perform these studies? I guess than in stability chambers.
  • Why did the authors perform temperature dependent studies at 4 instead of 5°C? Usually is 5 ± 3 °C. Include the standard deviation of the temperature ( I guess 4°C ± 3 and 25±2 °C).
  • Stability studies: How many replicates were tested?

Results:

  • Did the author plan to include the information of Figure 2 in a table? I mean, this figure is a little messy.
  • Figure 3. Have the authors the data of THCA and CBDA?

Discussion should be improved.

References: They should be updated. Medicinal cannabis is more than a current research area, and many papers and reviews have been published in the last five years, instead of including papers published more than 10 years ago. I suggest including more recent references in both introduction and discussion.

Author Response

Thank you for your suggestions.

The authors provided to modify the manuscript.

Best regards,
Antonio D’Avolio, (BSc, MSc, SM)

Reviewer 2 Report

This paper deals with an in-depth comparison of the quantitative analysis of two bioactive compounds in galenic preparations currently used in the clinical pharmacies in Italy. The topic is of interest in the last period and the law requirements are strict. The analyses were performed according to the international guidelines and the results are robust. The authors carried out the experiments using modern instruments and equipment also determining the stbility of the product up to 27 days. I strongly recommend the approval for this research article. Just a typo in paragraph 4.5.2 (tree laboratories).

Author Response

Thank you for your comments,

the authors provided to modify the manuscript following your suggestion.

Best regards,
Antonio D’Avolio, (BSc, MSc, SM)

Reviewer 3 Report

This study is meaningful because harmonization of cannabinids is medically important. The validation procedures are smart and the manuscript is well described.

However, the format does not follow the guidelines (especially in the senction of the reference list) and requires minor revision. Correct the manuscript according to the attached file and the guidelines . https://www.mdpi.com/journal/pharmaceuticals/instructions#preparation

I have some questions. Answer the following questions.

1)  p. 4, line 25.  The manuscript describes "within 20% for at least 67% of the determinations". I feel that it is a big error that the obtained result of up to 33% showed a difference of 20% or more. Do the EMA guidlines state that this leves is reliable?

2) p. 9, line 23.  Why pure olive oil was used in the method? 

Author Response

Thank you for your suggestions.

The authors revise the manuscript following your comments.

Best regards,
Antonio D’Avolio, (BSc, MSc, SM)

Round 2

Reviewer 1 Report

The authors have improved the manuscript by adressing all the comments and suggestions. The manuscript deserves to be published in the current (revised) form.